# Metabolic plasticity can amplify ecosystem responses to global warming

Rebecca L. Kordas[1], Samraat Pawar [1], Dimitrios-Georgios Kontopoulos [2,3], Guy Woodward [1] & Eoin J. O'Gorman [4✉]

Organisms have the capacity to alter their physiological response to warming through acclimation or adaptation, but the consequence of this metabolic plasticity for energy flow through food webs is currently unknown, and a generalisable framework does not exist for modelling its ecosystem-level effects. Here, using temperature-controlled experiments on stream invertebrates from a natural thermal gradient, we show that the ability of organisms to raise their metabolic rate following chronic exposure to warming decreases with increasing body size. Chronic exposure to higher temperatures also increases the acute thermal sensitivity of whole-organismal metabolic rate, independent of body size. A mathematical model parameterised with these findings shows that metabolic plasticity could account for 60% higher ecosystem energy flux with just +2 °C of warming than a traditional model based on ecological metabolic theory. This could explain why long-term warming amplifies ecosystem respiration rates through time in recent mesocosm experiments, and highlights the need to embed metabolic plasticity in predictive models of global warming impacts on ecosystems.

[1] The Georgina Mace Centre for the Living Planet, Department of Life Sciences, Imperial College London, Silwood Park Campus, Buckhurst Road, Ascot, Berkshire SL5 7PY, UK. [2] LOEWE Centre for Translational Biodiversity Genomics, Senckenberganlage 25, 60325 Frankfurt, Germany. [3] Senckenberg Research Institute, Senckenberganlage 25, 60325 Frankfurt, Germany. [4] School of Life Sciences, University of Essex, Wivenhoe Park, Colchester CO4 3SQ, UK. ✉email: e.ogorman@essex.ac.uk

A crucial, but currently unresolved, question in climate change research is how the functioning of complex ecosystems will respond to long-term warming. This challenge has typically been approached using general relationships for the size- and temperature-dependence of species-level metabolic rates in ecosystem-scale mathematical models[1,2]. Most of these studies assume, based on the Metabolic Theory of Ecology (MTE), that whole-organism metabolic rate increases with size (typically measured as body mass) raised to the three-quarter power, and with temperature according to the Boltzmann–Arrhenius equation with an activation energy of ~0.65 eV[3]. The assumed universality of these specific values has been questioned more recently, however, given that they vary both within and across species[4,5]. Furthermore, species may have the capacity to alter their metabolic traits through acclimation, evolutionary adaptation or both[6–10]. We refer to this flexibility in species-level thermal responses henceforth as 'metabolic plasticity', which can be thought of simply as a group of similar organisms altering their metabolic rate in a similar way when the environment changes. Metabolic plasticity should ultimately have consequences for ecosystem functioning by altering energy flow through the food web[11], but evidence for such changes across species and trophic levels in natural systems is still lacking.

To address this knowledge gap, we measured oxygen consumption rates (a standard measure of metabolic rate[12]) of freshwater invertebrates in a large-scale natural warming experiment. Our study site, the Hengill catchment in Iceland[13–18], consists of multiple streams, each with a characteristic temperature regime (Supplementary Fig. 1), resulting from long-term geothermal heating of the underlying bedrock. The invertebrate populations in each stream have been exposed to a distinct thermal regime over many generations, and thus provide an ideal natural experiment for elucidating the effects of chronic warming on physiology across trophic levels in the food web. This space-for-time substitution allows us to glimpse the potential impacts of centennial-scale warming[19], with previous research in the Hengill system revealing effects of temperature on biodiversity[13,14], community structure[17,18] and ecosystem functioning[15,16].

Our central hypothesis was that metabolic plasticity consistently declines with body size, given that smaller organisms tend to have faster rates of acclimation[20] and adaptation[21]. To test this, we collected thousands of invertebrates from nine different streams spanning mean annual temperatures of 5–20 °C (Supplementary Figs. 1–2). Invertebrates are the dominant primary and secondary consumers in the streams[13], and therefore central to energy fluxes through the food web[17,18]. We measured the individual-level routine metabolic rate (i.e., organisms exhibited some activity[12]) under acute exposure (experiments lasting 10–60 min) to temperatures of 5, 10, 15, 20 or 25 °C in the laboratory (see Methods). This response of metabolic rate to short-term temperature change is a fundamental measure of organismal physiology and also governs the daily effects of temperature on energy fluxes through ecosystems[1]. Metabolic rate was only quantified for each individual at one acute temperature, after which we measured its body mass and confirmed its species identity *via* microscopy. After quality-control procedures (see Methods), this yielded data on 1359 individuals from 16 species representing 44 different populations (Supplementary Table 1). Note that these species were not found in every stream, but there were still multiple populations of each species, whereby a population is a unique species × stream combination.

## Results and discussion

We found that chronic exposure to warmer environments (i.e., stream temperatures) altered both the size- and (acute)

temperature-dependence of metabolic rate in two fundamental ways (Table 1a). First, higher temperatures reduced the allometric scaling exponent of metabolic rate because smaller organisms had a more elevated metabolism after long-term warming than larger ones (Fig. 1a, b). This key finding suggests that metabolic plasticity can be modelled mechanistically, irrespective of species identity, through its relationship with body mass. Second, chronic warming raised the activation energy (thermal sensitivity) of metabolic rate, whereby organisms from warmer streams had a more elevated metabolism under acute warming (Fig. 1c, d). Higher metabolic rates can provide organisms with greater scope for faster growth or improved performance through increased activity, foraging, and competitive ability[6,22,23]. However, there is also a greater energy cost, and higher metabolism becomes disadvantageous if resource supply cannot keep pace[23,24]. These changes in two fundamental features of thermal physiology across multiple species exposed to chronic warming indicate a degree of metabolic plasticity that has never been documented before. This may have significant implications for ecosystem-level responses to climate change, such as altering total energy flux through the food web, or the amount of carbon emitted to the atmosphere through ecosystem respiration. Supplementary Analyses indicated no influence of spatial autocorrelation on these results, i.e., warm and cold streams are sufficiently mixed in the landscape (Supplementary Fig. 3). In addition, the optimal model describing our data (Table 1a) remained the same after accounting for phylogenetic information, suggesting that both evolutionary and acclimatory processes play important roles in shaping metabolic plasticity (Supplementary Tables 2–3; Supplementary Figs. 4–5; see Supplementary Analyses).

We used a general mathematical model of ecosystem energy fluxes to explore the potential implications of these empirical findings[25]. We parameterised the model with previously published empirical biomass and dietary data[18] from the study system and physiological rates that incorporated either metabolic plasticity (our new finding; Table 1a) or fixed thermal responses (the classical MTE assumption; Table 1b; see Methods). Note that the size-dependent nature of the model makes it independent of species identity and enables it to span multiple trophic levels[26], which should make it generally applicable to other ecosystems. Our estimates of energy flux reflected measurements of ecosystem

**Table 1 Statistical output of models exploring the key drivers of metabolic rate.**

| Model | Parameter | Value | SE | t value | p value |
|---|---|---|---|---|---|
| (a) With plasticity | $I_0$ | −11.03 | 0.1558 | −70.80 | <0.001 |
| | $\ln(M)$ | 0.6307 | 0.0461 | 13.70 | <0.001 |
| | $T_A$ | 0.7217 | 0.0319 | 22.61 | <0.001 |
| | $T_C$ | −0.1709 | 0.0647 | −2.641 | 0.008 |
| | $\ln(M){:}T_C$ | −0.0741 | 0.0205 | −3.619 | <0.001 |
| | $T_A{:}T_C$ | 0.1124 | 0.0230 | 4.881 | <0.001 |
| (b) Without plasticity | $I_0$ | −11.00 | 0.1595 | −68.96 | <0.001 |
| | $\ln(M)$ | 0.6518 | 0.0481 | 13.54 | <0.001 |
| | $T_A$ | 0.7015 | 0.0282 | 24.88 | <0.001 |

The estimated coefficients (value) for size- and temperature-dependence parameters are shown with standard errors (SE), t values, and p values, obtained from linear mixed-effects models fitted to metabolic rate data on 44 invertebrate populations from nine streams of different temperature (Fig. 1). In both models, log metabolic rate [$\ln(I)$ in J h$^{-1}$] was the dependent variable and the random effects structure included a random intercept for species identity and random slopes for each of the main effects. (a) The most parsimonious model included an intercept [$\ln(I_0)$], main effects of log body mass [$\ln(M)$ in mg], acute temperature exposure [$T_A$ in K], and chronic temperature exposure [$T_C$ in K], and interactive effects of $T_C$ on $\ln(M)$ and $T_A$. (b) An alternative model without metabolic plasticity contains only an intercept and main effects for $\ln(M)$ and $T_A$, in line with the general MTE prediction of a universal size-scaling and activation energy (but with ΔAIC > 31 (see Supplementary Table 7), indicating significantly weaker explanatory power than the model with metabolic plasticity.

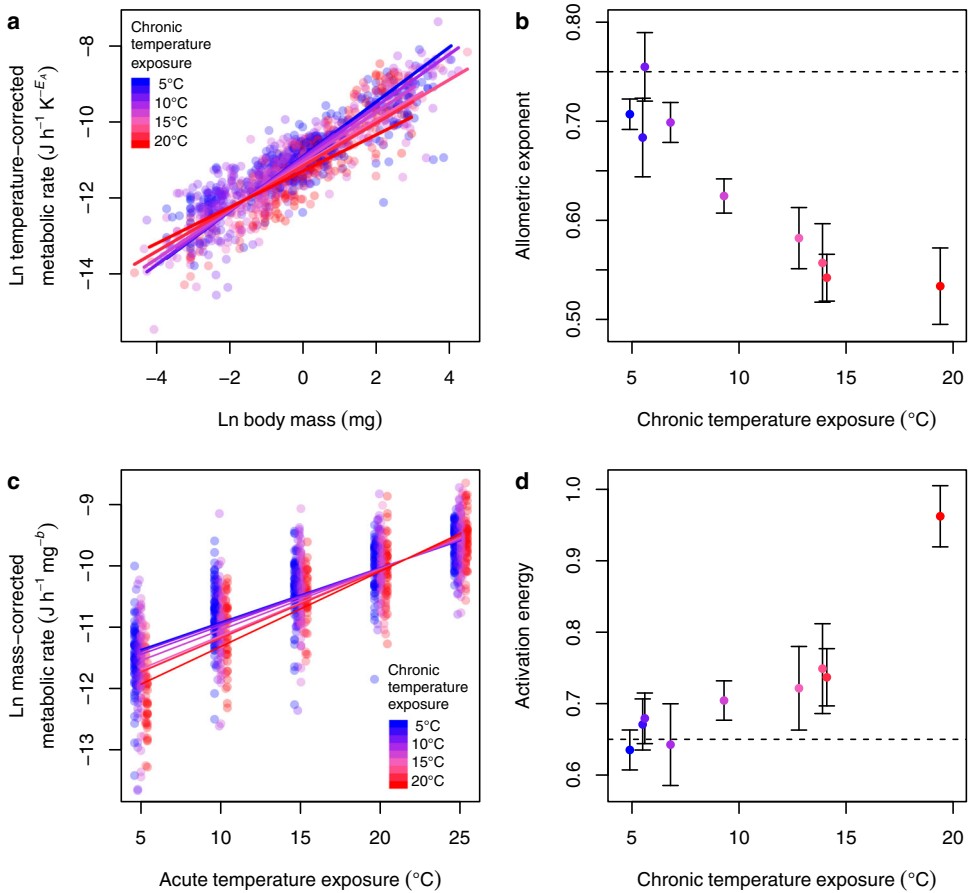

**Fig. 1 Chronic exposure to warmer conditions alters the size- and temperature-dependence of metabolic rate. a** Temperature-corrected metabolic rates are elevated for smaller organisms and suppressed for larger organisms after chronic exposure to warmer conditions, seen as (**b**) a decline in their allometric scaling exponent. The dashed line is the three-quarter scaling expected from MTE. **c** Mass-corrected metabolic rates are suppressed at lower acute temperatures and elevated at higher acute temperatures after chronic exposure to warmer conditions, seen as (**d**) an increase in their activation energy (thermal sensitivity). The dashed line is the typical activation energy of 0.65 eV expected from MTE for heterotrophic metabolism. The bars in (**b**) and (**d**) represent standard error around the mean from partial residual analysis of linear mixed-effects models fitted in (**a**) and (**c**), respectively (see Table 1a for model parameters and the Supplementary Note for underlying R code). Colours of points and lines in all panels indicate the environmental temperature to which species have been chronically exposed (see graphical legends). Source data are provided as a Source Data file.

respiration taken from the same streams[15] ($r^2 > 0.70$; Supplementary Fig. 6), showing that individual-level metabolic rate measurements can be used to predict ecosystem functioning.

Next, we used both models to estimate the change in energy flux under a global warming scenario of $+2\,°C$, predicted for the end of the century under intermediate IPCC scenarios of greenhouse gas emissions[19]. We expected higher energy flux in the model with metabolic plasticity due to the greater scope for elevated metabolism following long-term warming by smaller organisms near the base of the food web (Fig. 1). We found that warming always increased energy flux from resources to consumers, and this was greater by $59 \pm 9\%$ (mean ± standard error) for the model that included metabolic plasticity (Wilcoxon test: $V = 12$, $p = 0.004$; Fig. 2a, b). This was largely driven by increased energy flux from primary producers to herbivores (Wilcoxon test: $V = 11$, $p = 0.003$; Fig. 2c), with no significant differences between the models for detritivorous (Wilcoxon test: $V = 33$, $p = 0.121$; Fig. 2d) or predatory fluxes (Wilcoxon test: $V = 55$, $p = 0.572$; Fig. 2e).

These results indicate that current predictive models that ignore metabolic plasticity may substantially underestimate the changes in ecosystem fluxes under future warming. The associated increase in respiratory losses could help to reconcile the apparent paradox of amplified ecosystem respiration after 7 years

of warming in a pond mesocosm experiment, relative to the effects after just 1 year of warming[27]. If organisms were adapting to mitigate the effects of warming over longer timescales, then ecosystem respiration should converge with the controls in this experiment. The amplification of ecosystem respiration is thus surprising and should lead to greater carbon emissions to the atmosphere[27]. Our model predictions for amplified ecosystem fluxes following chronic exposure to warmer conditions thus highlight the value of using size-dependent metabolic plasticity to predict these surprising long-term warming effects on ecosystems. More generally, accounting for different rates of metabolic plasticity among species could provide a better understanding of how species-specific responses to global warming may sum up to alter ecosystem functioning.

To the best of our knowledge, this is the first study to identify consistent effects of chronic warming on the allometric scaling and activation energy of metabolic traits across trophic levels, and determining the mechanistic basis is an important area for future research. It is important to note that our study system is near the Arctic region, where organisms are more likely to be energetically constrained by the colder environment, and thus have greater scope for elevating their metabolism to increase foraging and growth rates as temperature increases (i.e., maximising their energy gain). In contrast, tropical organisms may find it more beneficial to

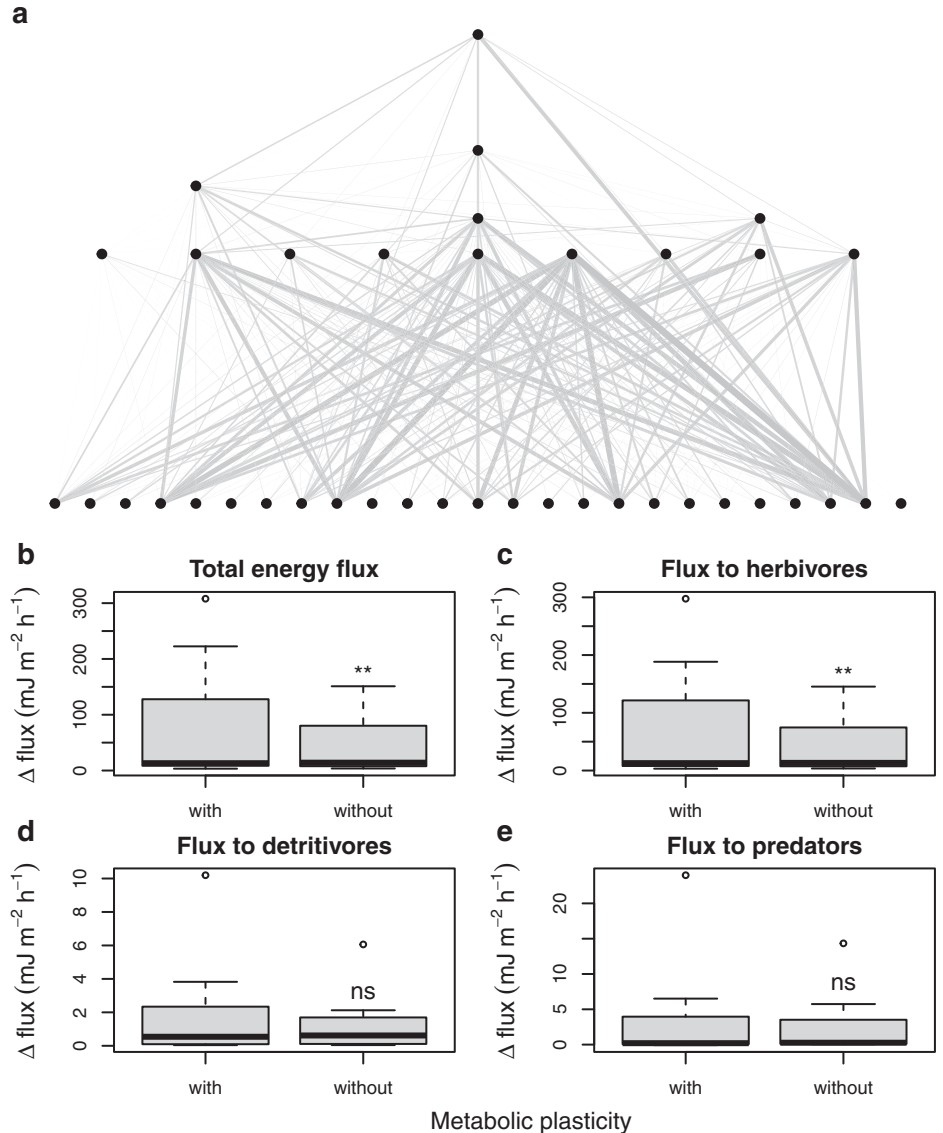

**Fig. 2 Effect of metabolic plasticity on the predicted impact of global warming on ecosystem-wide energy fluxes. a** Visualisation of changes in modelled energy fluxes through a food web in the study system after a simulated +2 °C of warming. Thicker lines (trophic links) between species populations (nodes) indicate a greater predicted increase in energy flux using the model with metabolic plasticity compared to the one without. **b** Using empirical data from our 14 study sites, the model with metabolic plasticity predicts a significantly greater increase in total energy flux through the networks following +2 °C of warming than the model without metabolic plasticity. This increase is driven by (**c**) flux to herbivores, with no significant contributions of increasing flux to (**d**) detritivores or (**e**) predators. Tukey box and whisker plots are shown, with the thick black line as the median, limits as the 1st and 3rd quartiles, whiskers as 1.5 times the interquartile range, and outliers as individual points. One-sided paired Wilcoxon tests were performed on the data in panels (**b–e**); $n = 14$ for every individual boxplot; \*\*$p \leq 0.01$; ns no significant difference. Source data are provided as a Source Data file.

downregulate their metabolism to minimise energy loss and avoid heat stress[28], highlighting the potential for local climate to interact with metabolic plasticity. Our findings may thus be most relevant for high latitude ecosystems, with temperate and tropical comparisons a priority for further studies on the topic. Our modelling framework should also be tested against ecosystems with different distributions of body sizes and trophic interactions to test the generality of our findings beyond our focal study system.

The levels of metabolic plasticity we report here (Fig. 1) are likely due to a combination of acclimation to warmer conditions (non-heritable changes within an organism's lifetime) and evolutionary adaptation (heritable changes over many generations). We did not empirically quantify the relative contribution of these two types of change, which is unfeasible to disentangle for multiple species in complex natural ecosystems. However, an

exploration of phylogenetic structure in our data suggests that evolutionary processes explain approximately half of the variability in metabolic rate (Supplementary Fig. 4). Most of the organisms in our study (13 of 16 species; see Supplementary Table 1) are larval invertebrates with an aerial adult phase, so genetic mixing between populations of the same species across streams is likely given that all the streams lie within 2 km of each other[14]. This suggests that thermal adaptation plays a weaker role in driving metabolic plasticity in our space-for-time study than might occur in response to long-term warming. Our results may thus be a conservative estimate of future change if genetic adaptation further increases the scope for metabolic plasticity over time.

Activity levels can be an important source of variation in estimates of temperature effects on metabolic rate[29]. Organisms

in our experiments were confined to small glass vials, and thus their activity was constrained during the experiments compared to normal activity. As such, our measurements are much closer to resting than maximum metabolic rate, and temperature effects are thus likely to indicate differences in energy allocated to growth, rather than activity. Nevertheless, quantification of activity levels and/or measurement of resting and maximum metabolic rates would be needed to disentangle the relative contributions of behavioural and physiological plasticity to the observed changes in rates of energy expenditure. Follow-up studies should prioritise this research gap.

Our findings have important implications for top predators, which are widely predicted to decline due to global warming[11,30]. Previous research in our study streams has shown that this is not necessarily the case and that larger organisms can thrive in warmer conditions if the production of resources is sufficient to meet their higher energy demands[16,17]. Thus, the scope for metabolic plasticity to increase energy flux from the base of the food web could help sustain large predators at higher trophic levels in the face of global warming. Understanding all these effects will require timescales of observation sufficient to disentangle adaptation from acclimation *via* decadal-scale warming experiments[11,31] and meta-analyses of time series data that incorporate long-term temperature changes[32]. In the meantime, embedding metabolic plasticity in individual-to-ecosystem projections of climate change impacts may improve realism and help to account for some of the ecological surprises in response to warming that have been reported recently[11,16,17,27].

## Methods

**Study system & organisms**. The study was conducted in the Hengill valley, Iceland[13–18] (N 64°03; W 21°18), which contains many streams of different temperature due to geothermal heating of the bedrock or soils surrounding the springs (Supplementary Fig. 1). The streams have been heated in this way for centuries[33] and are otherwise similar in their physical and chemical properties[13,18], providing an ideal space-for-time substitution in which to measure species responses after chronic exposure to different temperatures[6,34]. Fieldwork was performed in the summers of 2015–2018, between May and July. Stream temperatures were logged every 4 h using Maxim Integrated DS1921G Thermochron iButtons submerged in each stream (Supplementary Fig. 2). The average stream temperature over this study period was used as a measure of chronic temperature exposure, encompassing at least the lifetime of every invertebrate species under investigation (and potentially multiple generations[6,35].

Invertebrates were collected from nine streams spanning a temperature gradient of 5–20 °C across the entire study system (Supplementary Figs. 1–2). The streams exhibit some differences in the annual variability of their thermal regimes, but there are examples of both cold and warm streams that have high (IS12 and IS2) and low (IS13 and IS8) variability throughout the year. Our main finding is also robust to the inclusion of stream temperature variability as a random effect in our modelling framework (Supplementary Table 4; Supplementary Fig. 7). Note that we present temperature data from 15 streams in Supplementary Fig. 2, but it was not logistically feasible to study acute thermal responses of invertebrates collected from all of them, thus we focused on a subset of nine streams that best spanned the temperature gradient. The remaining six streams were included in other studies from the system, quantifying the biomass of all the constituent species[17], describing food web structure[18], and measuring whole-stream respiration[15] (described in detail below).

Individual organisms were stored in containers within their 'home stream' until the end of each collection day, when they were transported within 1 h to the University of Iceland and then transferred into 2 L aquaria filled with water from the main river in Hengill, the Hengladalsá. The water was passed through a 125 μm sieve to ensure no organisms or filamentous algae entered the aquaria, and thus limiting the potential food available to the study organisms. The aquaria were continuously aerated in temperature-controlled chambers set to the home stream temperature of the organisms during sampling, which were maintained without food for at least 24 h to standardise their digestive state prior to metabolic measurements[36]. While we did not observe any cannibalism or organisms feeding on dead bodies in the laboratory, we cannot rule out the possibility that organisms fed on fine algal or detrital particles in the water, thus increasing variability in our metabolic measurements due to differences in digestive state.

**Quantifying metabolic rates**. Experiments were carried out to determine the effects of body mass, acute temperature exposure (5, 10, 15, 20 and 25 °C), and chronic temperature exposure (i.e., average stream temperature) on oxygen consumption rates as a measure of metabolic rate[3,12]. Before each experiment, individual organisms were confined in glass chambers in a temperature-controlled water bath and slowly adjusted to the (acute) experimental temperature over a 15 min period to avoid a shock response. Glass chambers ranged in volume from 0.8–5 ml and scaled with the size of the organism. The glass chambers were filled with water from the Hengladalsá, which was filtered through a 0.45 μm Whatman membrane after aeration to 100% oxygen saturation. A magnetic stir bar was placed at the bottom of each chamber and separated from the organism by a mesh screen. In each experiment, one individual organism was placed in each of seven chambers and the eighth chamber was used as an animal-free control to correct for potential sensor drift. The chambers were sealed with gas-tight stoppers after the 15 min acclimatisation period, ensuring there was no headspace or air bubbles.

Oxygen consumption by individual organisms was measured using an oxygen microelectrode (MicroRespiration, Unisense, Denmark), fitted through a capillary in the gas-tight stopper of each chamber[37]. A total of 330 s measurement periods were recorded for each individual, where dissolved oxygen was measured every second. Oxygen consumption rate was calculated as the slope of the linear regression through all the data points from a single chamber, corrected for differences in chamber volume and the background rate measured from the control chamber (which was never >5% of the measured metabolic rates). We converted the units of this rate (μmol $O_2$ h$^{-1}$) to energetic equivalents (J h$^{-1}$) using atomic weight (1 mol $O_2$ = 31.9988 g), density (1.429 g L$^{-1}$), and a standard conversion[38] (1 ml $O_2$ = 20.1 J). Organisms generally exhibited some activity during experiments, thus these measurements can be classified as routine metabolic rates[12], which are more reflective of energy expenditure in field conditions. Nevertheless, activity levels were minimal due to the space constraints of the chambers (volume equal to 5–100 times the mass of the measured organism), indicating that the measured rates were likely to be closer to resting metabolic rates. Oxygen concentrations were never allowed to decline below 70% to minimise stress and avoid oxygen limitation. The system was cleaned with bleach at the end of each measurement day to avoid accumulation of microbial organisms on the insides of glass chambers and the water bath. In total, oxygen consumption rates were measured for 1819 individuals, none of which were ever reused in another experiment, thus every data point in the analysis corresponds to a single new individual (see below for details of how this dataset was curated to the final analysed subset of 1359 individuals based on quality-control procedures).

Following each experiment, individuals were preserved in 70% ethanol and later identified to species level under a dissecting microscope, except for Chironomidae, which were identified by examining head capsules under a compound microscope[39]. A linear dimension was precisely measured for every individual using an eyepiece graticule and converted to dry body mass using established length-weight relationships (Supplementary Table 1).

**Statistical analysis**. All statistical analyses were conducted in R 4.0.2 (see the Supplementary Note for full details of statistical R code). According to the Metabolic Theory of Ecology[3] (MTE), metabolic rate, $I$, depends on body mass and temperature as:

$$I = I_0 M^b e^{E_A T_A}, \tag{1}$$

where $I_0$ is the intercept, $M$ is dry body mass (mg), $b$ is an allometric exponent, $E_A$ is the activation energy (eV), and $T_A$ is a standardised Arrhenius temperature:

$$T_A = \frac{T_{acute} - T_0}{k T_{acute} T_0}. \tag{2}$$

Here, $T_{acute}$ is an acute temperature exposure (K), $T_0$ sets the intercept of the relationship at 283.15 K (i.e., 10 °C), and $k$ is the Boltzmann constant ($8.618 \times 10^{-5}$ eV K$^{-1}$). We performed a multiple linear regression ('lm' function in the 'stats' package) on the natural logarithm of Eq. (1) to explore the main effects of temperature and body mass on the metabolic rate of each population (i.e., species × stream combination) in our dataset[3]. Following these analyses, we excluded populations where $n < 10$ individuals, $r^2 < 0.5$, and $p > 0.05$ for any term in the model (see Supplementary Table 5, Supplementary Figs. 8–9). This excluded any poor quality species-level data and resulted in 1359 individuals in 44 populations for further analysis. Note that we find the same overall conclusion if we analyse the entire dataset (Supplementary Table 6, Supplementary Fig. 10).

To determine whether chronic temperature exposure alters the size- and acute temperature-dependence of metabolic rate, we added a term for chronic temperature exposure to Eq. 1. We began our analysis by considering the natural logarithm of all possible combinations of the main and interactive effects in this model:

$$lnI = ln I_0 + bln M + E_A T_A + E_C T_C + b_A ln M T_A + b_C ln M T_C + E_{AC} T_A T_C + b_{AC} ln M T_A T_C. \tag{3}$$

Here, $T_C$ is a standardised Arrhenius temperature with $T_{chronic}$ as a chronic temperature exposure (K) substituted for $T_{acute}$ in Eq. (2). To determine the optimal random effects structure for this model, we compared a generalised least squares model of Eq. 3 with linear mixed-effects models ('gls' and 'lme' functions in the 'nlme' package) containing all possible subsets of the following random effects

structure[40]:

$$random = \sim 1 + ln\,M + T_A + T_C | species. \qquad (4)$$

Here, we are accounting for the possibility that metabolic rate could be different for each species (i.e., a random intercept) and that the effect of body mass, acute temperature exposure, or chronic temperature exposure on metabolic rate could also be different for each species (i.e., random slopes).

The full random structure (Eq. 4) was identified as the best model using Akaike Information Criterion (ΔAIC > 31.2; see Supplementary Table 7). We used this random structure in subsequent analyses, set 'method = 'ML' in the 'lme' function, and performed AIC comparison on all possible combinations of the fixed-effect structure[40] (i.e., Equation 3). The optimal model was identified as follows:

$$ln\,I = ln\,I_0 + b\,ln\,M + E_A T_A + E_C T_C + b_C ln\,M T_C + E_{AC} T_A T_C. \qquad (5)$$

Note that while the model with an additional interaction between ln(M) and $T_A$ performed similarly (ΔAIC = 0.2; see Supplementary Table 8), that term was not significant ($t = -1.645$; $p = 0.1002$). We set 'method = 'REML' before extracting model summaries and partial residuals from the best-fitting model[40]. Note that the models were always fitted to the raw metabolic rate data, with residuals only extracted for a visual representation of the best-fitting models, excluding the noise explained by the random effect of species identity (see R code in the Supplementary Note).

**Exploration of spatial autocorrelation.** A Mantel test ('mantel' function in the 'vegan' R package) was used to test for spatial autocorrelation in the temperature gradient, by comparing pairwise temperature difference between streams to the pairwise distance between streams. Pairwise distances were calculated from GPS coordinates taken at the confluence of each stream with the main river and the 'earth.dist' function in the 'fossil' R package. This analysis revealed no significant relationship between pairwise temperature and pairwise distance between sites (Mantel $r = -0.1293$, $p = 0.780$).

In addition, we explored for spatial autocorrelation in the residuals of our optimal model (Table 1a) by generating an empirical semivariogram cloud, illustrating the squared difference between all pairwise residual data points as a function of the distance between the two points. We also calculated Moran's I as a measure of spatial autocorrelation in the model residuals. The semivariogram indicated no clear patterns in the residuals as a function of the distance between data points (Supplementary Fig. 3) and there was no statistical evidence for spatial autocorrelation in the model residuals (Moran's $I = 0.1187$, $p = 0.453$).

**Exploration of phylogenetic structure.** To examine the influence of evolutionary relatedness on metabolic rate measurements, we reconstructed a time-calibrated phylogeny of the 16 species in our final dataset (Supplementary Table 1). To this end, we combined: (i) nucleotide sequences of the 5′ region of the cytochrome c oxidase subunit I gene (COI-5P) from the Barcode of Life Data System database[41]; (ii) tree topology information from the Open Tree of Life[42] (OTL; v. 13.4); and (iii) previously reported divergence time estimates between pairs of genera from the TimeTree database[43]. More precisely, we were able to obtain COI-5P nucleotide sequences for 15 out of 16 species (Supplementary Table 2), which we aligned using the G-INS-i algorithm of MAFFT[44] (v. 7.490). To constrain the topology of our phylogeny based on the results of previous studies, we queried the OTL via the 'rotl' R package[45] (v. 3.0.11). This yielded topological information for all 16 species. Finally, we manually queried the TimeTree database to obtain node age estimates. We only used three such estimates that (a) were based on more than five previous studies and (b) did not force any tree branches to have a length of zero.

We next used MrBayes[46] (v. 3.2.7a) to obtain a time-calibrated phylogeny based on the sequence alignment, the OTL topology, and the node ages from TimeTree. For this, we first determined the most appropriate nucleotide substitution model using ModelTest-NG[47] (v. 0.1.7). This was the General Time-Reversible model with Gamma-distributed rate variation across sites and a proportion of invariant sites. To allow branches of the phylogeny to differ in their rate of sequence evolution, we specified the Independent Gamma Rates model[48] and used a normal distribution with a mean of 0.00003 and a standard deviation of 0.00001 as the prior for the mean clock rate. Finally, we executed four MrBayes runs with two chains per run for 100 million generations, sampling from the posterior distribution every 500 generations. Samples from the first ten million generations were treated as burn-in and were discarded. We examined the remaining samples to ensure that the four MrBayes runs had converged on statistically indistinguishable posterior distributions (i.e., all potential scale reduction factor values were below 1.1) and the parameter space was sufficiently explored (i.e., all effective sample size values were higher than 200). We summarised the sampled trees into a single time-calibrated phylogeny by calculating the median age estimate for each node (Supplementary Fig. 4).

To investigate the influence of evolutionary and acclimatory processes on metabolic rate, we first estimated the phylogenetic heritability of metabolic rate, i.e., the extent to which closely related species have more similar trait values than species chosen at random[49]. This metric takes values from 0 (trait values are independent of the phylogeny) to 1 (trait values evolve similarly to a random walk in the parameter space), with intermediate values indicating deviations from a pure random walk. To estimate phylogenetic heritability, we fitted a generalised linear mixed-effects model using the 'MCMCglmm' R package[50] (v. 2.32). We set the

natural logarithm of metabolic rate as the response variable and only an intercept as a fixed effect. We also specified a phylogenetic species-level random effect on the intercept, using the phylogenetic variance-covariance matrix obtained from our time-calibrated phylogeny. We used the default (normal) prior for the fixed effect, an uninformative Cauchy prior for the random effect, and an uninformative inverse Gamma prior for the residual variance. We then executed four independent runs for 500,000 MCMC generations each, with parameter samples being obtained every 50 generations after the first 50,000. We verified that sufficient convergence was reached, based on potential scale reduction factor and effective sample size values, as described earlier. Phylogenetic heritability was calculated as the ratio of the variance captured by the species-level random effect to the sum of the random and residual variances. The mean posterior phylogenetic heritability estimate of the natural logarithm of metabolic rate was 0.48. This means that nearly half (48%) of the variation can be explained by the evolution of metabolic rate along the phylogeny (Supplementary Fig. 4), with the other half arising from other sources including (but not necessarily limited to) acclimation and measurement error.

To describe the remaining unexplained variation, we fitted a series of models using MCMCglmm in R with all possible combinations of log body mass, acute temperature exposure, and chronic temperature exposure (fixed effects, as in Eq. 3) of the main text) and species-level random effects on the intercept and slopes (as in Eq. 4) of the main text). Furthermore, we specified both phylogenetic and non-phylogenetic variants of each model to understand if such a correction is warranted when the fixed effects are included. We determined the most appropriate model based on the Deviance Information Criterion[51] (DIC). The optimal model (ΔDIC > 19; Supplementary Table 3; Supplementary Fig. 5) was found to include the full random effects structure (Eq. 4), the main effects of log body mass, acute temperature exposure, and chronic temperature exposure, the interaction between log body mass and chronic temperature exposure, and the interaction between acute temperature exposure and chronic temperature exposure (as for Eq. 5 in the main text), i.e., the same optimal model as that containing only species-level, rather than phylogenetic, information (Table 1a; Fig. 1). We calculated the marginal and conditional coefficients of determination to report the amounts of variance explained by the fixed and random effects, or left unexplained[52]. We found that the unexplained variation dropped from 52% to 8%, indicating that metabolic rate is strongly influenced by acclimatory processes in addition to evolutionary processes (see above).

It should be noted, however, that a definitive empirical quantification of the relative strength of evolutionary and acclimatory processes would require population genetics (to determine evolutionary divergent populations among streams), transcriptomics (to identify the expression of genes associated with thermal adaptation), and exhaustive common garden experiments (to disentangle acclimation from adaptation in all populations). Such an undertaking was logistically unfeasible in this study, but should be a focus for follow-up research on this topic.

**Modelling ecosystem-level energy fluxes.** We used a recently proposed approach for inferring energy fluxes through trophic links[25] to predict the effects of climate warming on ecosystem-level energy fluxes. We began by assuming that each stream ecosystem is at energetic steady state, i.e., for all $n$ consumer species in the system:

$$G_i = L_i, \; i = 1, 2, \ldots, n, \qquad (6)$$

where $G_i$ and $L_i$ are the energy gain and loss rates [J h$^{-1}$], respectively, of the $i$th species in that stream. All basal species are implicitly assumed to be at energy balance. The two terms in Eq. (6) can be specified in a general way as

$$G_i = \sum_{k \in R_i} e_{ki} w_{ki} F_{ki}, \text{ and} \qquad (7)$$

$$L_i = Z_i + \sum_{j \in C_i} w_{ij} F_{ij}. \qquad (8)$$

Here, for the $i$th species, $R_i$ and $C_i$ are the sets of its resource and consumer species respectively, and $Z_i$ is its population-level energy loss rate stemming from mortality and metabolic expenditure on various activities realised over the timescale of the system's dynamics. For the $j$th species feeding on the $i$th species, $F_{ij}$ is the maximum population-level feeding rate, $e_{ij}$ is the assimilation efficiency (expressed as a proportion), and $w_{ij}$ is the consumer's preference for that species (all preferences for a given consumer sum to 1). Thus, the effective flux through a trophic link is $e_{ki} w_{ki} F_{ki}$. Next, assuming the energy balance condition in Eq. 6 holds for all species, there are $n$ linear equations (corresponding to the $n$ consumer species) of the form:

$$G_i - L_i = \sum_{k \in R_i} e_{ki} w_{ki} F_{ki} - \left( Z_i + \sum_{j \in C_i} w_{ij} F_{ij} \right) = 0, \qquad (9)$$

which can be solved iteratively to obtain the unknown fluxes $F_{ij,i \neq j}$ of all consumer species, provided all the $Z_i$'s, $e_{ij}$'s, and $w_{ij}$'s are known.

For this, we used the 'fluxing' function in the 'fluxweb' R package, parameterised with: (1) binary predation matrices for 14 stream food webs, characterised by 49,324 directly observed feeding interactions[18]; (2) biomasses for every species in each food web, characterised by 13,185 individual body mass measurements[17]; (3) assimilation efficiencies ($e_{ij}$'s) based on an established temperature-dependence and resource type (i.e., plant, detritus, or invertebrate)[53]; (4) preferences ($w_{ij}$'s)

depending on resource biomasses; and (5) metabolic rates estimated using Eqs. (1) and (5) (assuming that $I$ approximates $Z$). We treated $T_A$ in Eqs. (1) and (5) as the short-term temperature of the streams during food web sampling[17,18] and $T_C$ in Eq. (5) as the long-term average temperature of the streams measured over the current study (Supplementary Fig. 2). It is important to note that the energy balance assumption (Eq. 6) implies that $Z_i$ in Eq. (8) is a combination of basal, routine, and active metabolic rates, stemming from the combination of activities realised over the timescale of the system's dynamics. Therefore, our use of routine metabolic rate $I$ is an underestimate of $Z$, which in turn means that the fluxes (which must balance the losses) are an underestimate.

Biomass and food web data were sampled in August 2008, with extensive protocols described in previous publications[17,18]. Briefly, this involved three stone scrapes per stream for benthic diatoms, five Surber samples per stream for macroinvertebrates, and three-run depletion electrofishing for fish. All individuals in the samples were identified to species level where possible and counted. Linear dimensions were measured for at least ten individuals of each species in each stream, with body masses estimated from length-weight relationships[17]. The population biomass of each species in each stream was calculated as the total abundance [individuals m$^{-2}$] multiplied by the mean body mass [mg dry weight]. Food web links were largely assembled from gut content analysis of individual organisms collected from the streams (>87% of all links in the database), but additional links were added from the literature when yield-effort curves indicated that the diet of a consumer species was incomplete[18].

**Validation of the ecosystem flux model using field data**. To test whether our model of energy fluxes through trophic links was empirically meaningful, we calculated the sum of all energy fluxes through each stream food web to get the total energy flux, $F$ (i.e., the sum of all $e_{ki}w_{ki}F_{ki}$'s in Eq. 7). This quantity is a measure of multitrophic functioning and is expected to be positively correlated with the total respiration of each stream[25]. To evaluate this, we compared $F$ to whole-ecosystem respiration rates measured in the same study streams[15]. The ecosystem respiration estimates were based on a modified open-system oxygen change method using two stations corrected for lateral inflows[54,55]. Essentially, this was an in-stream mass balance of oxygen inflows and outflows along stream reaches (17–51 m long). Oxygen concentrations were measured during 24- to 48 h periods from 6th to 16th August 2008, i.e., the exact same time period during which biomass and food web data were sampled to parameterise the energy flux model[15]. Dissolved oxygen concentrations were measured every minute with optic oxygen sensors (TROLL9500 Professional, In-Situ Inc. and Universal Controller SC100, Hach Lange GMBF). Hourly ecosystem respiration was calculated from the net metabolism at night, i.e., when no primary production occurs due to lack of sunlight.

**Modelling the consequences of metabolic plasticity for global warming impacts on ecosystem-level energy flux**. In addition to total energy flux, $F$, we also calculated a modified total energy flux, $F^*$, for each food web after considering a global warming scenario, where we added 2 °C to $T_A$ in Eq. (1) and to both $T_A$ and $T_C$ in Eq. (5). We calculated the change in total energy flux as a result of the global warming scenario as $\Delta F = F^* - F$. We tested whether the (statistically optimal) model with metabolic plasticity (Eq. 5) predicted a greater $\Delta F$ across the 14 empirical stream food webs from the Hengill system than the model without metabolic plasticity using paired Wilcoxon tests (since the data did not conform to homogeneity of variance). To determine whether our results were consistent for all major trophic groupings in the system, we repeated the analysis after calculating the change in energy flux to herbivores ($\Delta F_H = F_H^* - F_H$), detritivores ($\Delta F_D = F_D^* - F_D$), and predators ($\Delta F_P = F_P^* - F_P$) in each stream.

**Reporting summary**. Further information on research design is available in the Nature Research Reporting Summary linked to this article.

## Data availability
The data generated in this study have been deposited with the University of Essex Research Data Repository at https://doi.org/10.5526/ERDR-00000148. Source data for all figures are also provided with this paper.

## Code availability
The R code generated in this study has been deposited with the University of Essex Research Data Repository at https://doi.org/10.5526/ERDR-00000148.

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

## Acknowledgements

We thank Gísli Már Gíslason and Jón S. Ólafsson for providing valuable advice, research support, and laboratory facilities, and Benoit Demars for additional whole-ecosystem respiration data. We thank Luis Moliner Cachazo, Filipa Esteves Goncalves, and Joana Neto-Cerejeira for help with sample processing and respiration experiments. We acknowledge funding from NERC (NE/L011840/1 [EJOG], NE/M020843/1 [GW; SP; RLK; EJOG]), the Royal Society (RG140601 [EJOG]), and NSF (PRFB 1401656 [RLK]).

## Author contributions

R.L.K., E.J.O.G., S.P. and G.W. were responsible for funding application and research design. R.L.K. and E.J.O.G. collected the data. E.J.O.G. and S.P. analysed the data. D.G.K. performed the phylogenetic analysis. E.J.O.G. wrote the first draft, with editorial input from all authors.

## Competing interests

The authors declare no competing interests.
