## [Peer Review File · Nature Communications]

Reviewers' Comments::

Reviewer #1:

Remarks to the Author:

This is a very compelling piece of research focusing on the consequences of variation in metabolic rate caused by temperature to energy flux as a core ecosystem service.

It is framed in the context of metabolic theory that ties metabolic rate to body size and metabolic rate to temperature. The MS presents data that indicate that "the ability of organisms to raise their metabolic rate following chronic exposure to warming decreases with body size" and that the "chronic warming also increases sensitivity to short term warming, irrespective of body size". Via a mathematical model specific to this system, these data demonstrating 'metabolic plasticity' are used to show that this variation can deliver an 'additional 60%' of energy flux.

The two most significant contributions appear to be

- a) there is metabolic plasticity mediated by body size and temperature
- b) this plasticity leads to additional variation in energy flux, a key ecosystem service, and offers an explanation to why ecosystem respiration in experiments continues to rise over longer periods than expected.

There are four spaces where the MS might benefit from more nuanced arguments/analyses.

First, details on the taxonomy and structure of the sampled organisms and the associated statistical modelling is not clear enough. The main text details 1359 individuals from 44 populations after 'cleaning'. The 'in MS' methods detail 1819 individuals. 15 streams are identified in the supplement, but the text indicates 20 invertebrate species from nine different streams. The text also indicates that populations are defined by species x stream combinations, which further confuses things. Thus, there appears to be very little coherence about the samples/species/populations/stream.

This is rather important, as the analysis requires attention to the spatial and taxonomic detail in order to isolate the temperature and body size effects.

Currently, the statistical modelling treats species as the sole random effect. It is likely that populations within the same streams, or even between streams and despite temperature differences, share a geographic/spatial autocorrelation. Demonstrating that this spatial component to the data is not an issue, or capturing it, is probably necessary.

Furthermore, it is perhaps now best practice to perform analyses of allometric relationships in the context of phylogenetic information, not just species identity. Given that the authors have full details on the genus and species of the 20 invertebrates, one might expect full use of a phylogeny (easily crafted from databases). Even if this were not possible, inclusion of a nested hierarchy of taxonomy would be a stop-gap.

Overall, the MS would benefit from a more robust statistical analyses that account for the spatial and phylogenetic structure of the data. In most analyses that do this, standard errors actually increase. One might wonder whether the results of the ANCOVA like analysis (e.g. quite subtle variation in the slopes, Fig 1) will be retained once these sources of variation are accounted for.

Second, the introduction to the work highlights two sources of 'plasticity' - acclimation, which I interpret from the writing as short term (non-evolutionary) physiological responses of the different individuals to temperature variation - and adaptation, which I interpret from the writing as evolutionary change derived from allele frequency changes.

It's probably worth noting that in most of evolutionary biology, sources of variation derived from adaptation are not considered plasticity, which is reserved for variation not attributable to allele frequency changes in a population. Thus, I expect readers will have trouble aligning the idea of metabolic plasticity with physiological adaptation. Given the nature of the work, question, and the

sampling and measuring over a very short term of individuals, one could probably just leave the idea of adaptation out of this.

However, it is worth recognising that perhaps the reason for bringing this idea into the MS is that the chronic exposure natural experiments are indicative of long term adaptation to a temperature regime. But these differences among chronic temperature are rarely defined as plasticity.

Perhaps there is opportunity, however, to consider more carefully how the 'levels of metabolic plasticity' could be due to a combination of acclimation or adaptation. Plasticity is defined by how a group (population, size class, species) responds to two or more different environments. That this plasticity might vary with the history the 'grouping variable' experiences is a very well established way to ask whether evolution has shaped how plastic populations are. It strikes me that the analysis that crosses chronic and acute conditions are indeed partitioning the impacts of adaptation to a temperature regime with acute exposure to a wider variety of conditions. Articulating this quantitatively does require careful attention to the statistical modelling.

Third, it is not clear in the MS just why evidence for metabolic plasticity and its effects on ecosystem function is so important to understand. The abstract leaves an impression that it is that some experiments have had community respiration rise in experiments even after treatments cease (the wording is a bit hard to understand...). Is this the major contribution of the findings in the MS?

It is compelling that the variation reporting in the MS can account for an additional 60% of energy flux, but it's not clear what this means: is the MS indicating that there was a lot of unexplained variation in energy flux in natural communities and that metabolic plasticity captures it? Or that including plasticity in the modelling leads to 60% higher energy flux than previously reported?

Overall, the MS would thus benefit from more clarity on the gap in knowledge that the data and results fill. Of course, another framing might be 'how does accommodating metabolic plasticity in metabolic rates among species (interspecific variation) alter our understanding of ecosystem function'. This is perhaps more generic and leaves open the options that the persistent averaging implicit in the core allometry of the MTE, used in a wide variety of circumstances, is just fine, or it isn't.

Finally, the MS would benefit from a gentle reminder about how narrow or general the mathematical model is. There are strong references to the development and use of the model in other papers and against other questions and some big numbers on feeding interactions and body masses thrown about. In the end, it's a well parameterised and well established model of the system. But what features of this make it general enough to then remark on the role of metabolic plasticity in ecosystem function? I imagine that readers may not be convinced that the model parameterised to this system (very classy and well done as it is) is the most effective tool for generalising about the effects of metabolic plasticity. If the consequences were to hold up in more general model formulations - e.g. across different distribution of body sizes and or interaction distributions - the message would be very strong indeed. Perhaps a bit more care is needed here to balance the incredible opportunity for accuracy of modelling this system with the capacity to generalise beyond it.

Reviewer #2:

Remarks to the Author:

In this paper, the authors measure metabolic rate of stream invertebrates across a range of species, body masses, temperatures, and 'home' climates to evaluate the plasticity of the size- and temperature-dependence of metabolic rate. They leverage the very cool Iceland stream system to do this, since it provides a natural opportunity to evaluate the consequences of long-term exposure to temperature change. The results suggest that, yes, the temperature you experience over long periods of time influences your response to short-term changes in temperature (metabolic plasticity). The results both contradict notions of universal temperature- and size-dependence of metabolic rate, and using a flux model, suggest that the observed

plasticity could have long-term effects for the consequences of warming on ecosystem respiration. Overall I enjoyed reading the study. It was well-written and clear, and directly explores an important question with a large and compelling data set. The methods appear sound to me. The results also should be highly impactful, as they provide a compelling update to broad views on the mass- and temperature-dependence of metabolic rate.

Some small points:

L49 – Previous research where? On this system?

L56 – It seems unlikely that the O₂ measurements are for 'normal' situations. Confined in a vial, these seem more likely to reflect resting metabolic rate rather than routine. If they could swim and flee predators and forage, maybe 'normal', but without that they should be thought of as closer to resting.

L68 – What do you mean by 'capable'? Are you suggesting that changes in metabolic rate are intentional rather than biochemical? Might want to rephrase or explain this point.

L147 – Do you actually expect warming to influence this system, since the temperature is controlled by the geothermal energy?

Reviewer #3:

Remarks to the Author:

Line 16-17: I think the statement "empirical evidence for this metabolic plasticity across species within food webs is lacking" is ambiguous as it could be interpreted as meaning that we know little about metabolic plasticity, which is not the case. There has been considerable research on metabolic plasticity. For instance, in ectothermic animals alone, Seebacher et al. (2015) collated 637 measurements of 202 species of the thermal acclimation responses of physiological traits including metabolic rates, heart rates, enzyme activities and locomotor performance. And presumably the animal species within this dataset occupy different levels within food webs. I would suggest rephrasing this statement to something like "the consequences of among-species variation in metabolic plasticity for energy flow through food webs is currently unknown".

Line 20-21: I think rephrasing this sentence to something like "Chronic exposure to higher temperatures also increases the acute thermal sensitivity of whole-organismal metabolic rate independent of body size" would improve clarity.

Line 36-37: I think Seebacher et al. (2015) should be cited here as well.

Line 54: The streams also seem to vary considerably in the extent to which they vary seasonally. For instance, IS10, IS13 and IS8 have relatively stable thermal regimes ranging from 3.4-19.4°C, whereas IS12 and IS1 show dramatic seasonal fluctuations in temperature. Have you considered including a measure of seasonal variability into your models to determine if this also contributes to variation in the acute thermal sensitivity of metabolic rate or to variation in the scaling of MR with body size? This would be particularly interesting in the context of the theory of metabolic cold adaptation (MCA), which suggests that organisms living in colder climates (such as those high latitudes and altitudes) should have higher metabolic rates than those living in warmer climates when measured at the same temperature. Some comparative analyses that examine how metabolic rate varies along geographic gradients (such as latitudinal clines) have found support for the MCA hypothesis (e.g. Addo-Bediako et al. 2002), but such analyses are unable to determine whether this observed pattern is the result of variation in mean annual temperature or variation in seasonality (see Alton et al. 2017).

Line 69-77: I think it is important that you provide some discussion about the limitations of the experimental component of your research as this would provide some valuable guidance for future research. In particular, I think it would be valuable if you acknowledged that you did not measure activity levels during your metabolic rate measurements and therefore cannot account for this important source of variation in estimates of metabolic rate (see Halsey et al. 2015). By not conducting simultaneous measures of activity and metabolic rate you cannot disentangle changes in activity from changes in mass-independent resting metabolic rates (which should provide a better estimate of the cost of self-maintenance), and thus you are unable to determine the relative contributions of behavioural plasticity and physiological plasticity to observed changes in rates of

energy expenditure. While you say that “higher metabolic rates can provide organisms with greater scope for faster growth”, this would only be the case if it was resting metabolic rates that increased (and only if resting metabolic rates are indicative of the capacity of individuals to process food and positively correlated with fitness as per the increased-intake hypothesis, see Nilsson 2002). If it was increases in activity levels that was causing routine metabolic rates to increase, then this would suggest that more energy was being allocated to activity, which may reduce the amount of energy available for growth. That you did not measure activity is not intended to be criticism – in my opinion, your measures of metabolic rate in this system are an incredibly valuable contribution to the field of research dedicated to understanding the influence of temperature on metabolic rates. I do however think that an acknowledgement of this limitation only adds value to your contribution because it should help to inspire future research directions.

Line 98, Table 1: Change “9 streams” to “nine streams”.

Line 112: Your experimental data provides a measure of metabolic plasticity in invertebrates that you say are the dominant primary and secondary consumers in the streams. Did you assume that the same level of plasticity across all trophic levels or just within the trophic levels that you have data for?

Line 268-269: Were these experiments conducted throughout the year or only during particular seasons?

Line 278-282: Was the water from the main river filtered? Was the home-stream temperature set to the average stream temperature or the temperature at the time of collection? How do you know that your organisms were not feeding while they were being held in the lab? If they were being housed together in groups, they may have eaten each other, or if individuals died then those that were living may have fed on the carcasses. It is entirely reasonable in such cases that you would not know the digestive states of the individuals and therefore just have to accept that this is source of variance in your data that you cannot account for.

Line 294: Is the acclimation period you mention here the same 15-minute period over which temperature was changed?

Line 298-299: You mention on Line 57 that measurements were for less than 1 h, but what was the minimum measurement duration?

Line 301-302: What were the chamber volumes? What was your cleaning protocols for the chambers and what were your calibration protocols for the oxygen sensors? I would highly recommend consulting Killen et al. (2021) who provide a checklist of criteria for reporting aquatic respirometry methods. While this checklist is for intermittent-flow respirometry, some of the criteria are also relevant for closed-system respirometry. This level of detail can be presented in the supplementary material.

Tables 1 and S3-S5: In your tables you have used the abbreviations of TS and TL but have not explained what these mean in the Table legends. Is TS stream temperature and thus equivalent to chronic temperature, TC? I do not know what TL would stand for?

References:

Addo-Bediako, A., Chown, S. L. & Gaston, K. J. (2002) Metabolic cold adaptation in insects: a large-scale perspective. *Functional Ecology*, 16, 332-338.

Alton, L. A., Condon, C., White, C. R. & Angilletta, M. J., Jr (2017) Colder environments did not select for a faster metabolism during experimental evolution of *Drosophila melanogaster*. *Evolution*, 71, 145-152. <http://dx.doi.org/10.1111/evo.13094>

Halsey, L.G., Matthews, P.G.D., Rezende, E.L. et al. The interactions between temperature and activity levels in driving metabolic rate: theory, with empirical validation from contrasting ectotherms. *Oecologia* 177, 1117–1129 (2015). <https://doi.org/10.1007/s00442-014-3190-5>

Nilsson, J.Å. (2002) Metabolic consequences of hard work. *Proc. R. Soc. Lond. B.* 269: 1735–1739
<http://doi.org/10.1098/rspb.2002.2071>

Shaun S. Killen, Emil A. F. Christensen, Daphne Cortese, Libor Závorka, Tommy Norin, Lucy Cotgrove, Amélie Crespel, Amelia Munson, Julie J. H. Nati, Magdalene Papatheodoulou, David J. McKenzie; Guidelines for reporting methods to estimate metabolic rates by aquatic intermittent-flow respirometry. *J Exp Biol* 15 September 2021; 224 (18): jeb242522. doi:
<https://doi.org/10.1242/jeb.242522>

*** Note that original reviewers' comments are in bold. Our responses are numbered sequentially, written in plain font, and contain quotations to the revised text in italics with line numbers corresponding to the revised version of the manuscript ***

Reviewer #1 (Remarks to the Author):

This is a very compelling piece of research focusing on the consequences of variation in metabolic rate caused by temperature to energy flux as a core ecosystem service. It is framed in the context of metabolic theory that ties metabolic rate to body size and metabolic rate to temperature. The MS presents data that indicate that ‘the ability of organisms to raise their metabolic rate following chronic exposure to warming decreases with body size’ and that the “chronic warming also increases sensitivity to short term warming, irrespective of body size”. Via a mathematical model specific to this system, these data demonstrating ‘metabolic plasticity’ are used to show that this variation can deliver an ‘additional 60%’ of energy flux.

Response #1: Thank you for the kind words and positive overall impression of our manuscript. We appreciate the constructive feedback you have given us below and hope we have satisfactorily addressed all of your points.

The two most significant contributions appear to be

a) there is metabolic plasticity mediated by body size and temperature

b) this plasticity leads to additional variation in energy flux, a key ecosystem service, and offers an explanation to why ecosystem respiration in experiments continues to rise over longer periods than expected.

Response #2: We agree with this assessment of the key contributions from our study.

There are four spaces where the MS might benefit from more nuanced arguments/analyses. First, details on the taxonomy and structure of the sampled organisms and the associated statistical modelling is not clear enough. The main text details 1359 individuals from 44 populations after ‘cleaning’. The ‘in MS’ methods detail 1819 individuals. 15 streams are identified in the supplement, but the text indicates 20 invertebrate species from nine different streams. The text also indicates that populations are defined by species x stream combinations, which further confuses things. Thus, there appears to be very little coherence about the samples/species/populations/stream. This is rather important, as the analysis requires attention to the spatial and taxonomic detail in order to isolate the temperature and body size effects.

Response #3: We were quite transparent in the original methods that we measured oxygen consumption rates for 1,819 individuals, but this number was curated down to a subset of 1,359 individuals for the final analysis. We believe this curation is very important to exclude poor quality data, which we defined in the original text as situations where we managed to collect fewer than 10 individuals for a given population, or where a multiple linear regression of log respiration rate as a function of log body mass and temperature produced an $r^2 < 0.5$ or $p > 0.05$ for any term in the model. It is also very important to note that we presented an analysis of the entire dataset of 1,819 individuals in our supplementary information for transparency, showing that our key results are unchanged (originally Fig. S6, now Fig. S10). However, we appreciate that it might be confusing for the reader to first encounter the statement in the extended Methods that we measured 1,819 individuals, when we mentioned in the main text that the final analysis was on 1,359 individuals. We now clearly state at first mention of 1,819

individuals in the revision that there is more information coming on how this larger dataset was curated down to the final subset based on quality-control procedures:

Ln380: *“In total, oxygen consumption rates were measured for 1,819 individuals, none of which were ever reused in another experiment, thus every data point in the analysis corresponds to a single new individual (see below for details of how this dataset was curated to the final analysed subset of 1,359 individuals based on quality-control procedures).”*

We also appreciate the confusion that there are more streams in the Hengill system ($n = 15$) than we could include in this study ($n = 9$). It was a huge amount of effort, spread over four years, to collect enough organisms from multiple streams of different temperature and acutely expose them to different experimental temperatures in the laboratory. For logistical reasons, we had to limit the number of streams we collected organisms from to just nine of the possible 15, choosing them to best span the available temperature gradient. We have added a paragraph to the revised text where we explain this, whilst noting that the other six streams were included in studies that quantified food web structure, biomass of organisms, and ecosystem respiration – measurements that are utilised later in the paper when we model energy fluxes:

Ln331: *“Note that we present temperature data from 15 streams in Fig. S2, but it was not logistically feasible to study acute thermal responses of invertebrates collected from all of them, thus we focused on a subset of nine streams that best spanned the temperature gradient. The remaining six streams were included in other studies from the system, quantifying the biomass of all the constituent species¹⁷, describing food web structure¹⁸, and measuring whole-stream respiration¹⁵ (described in detail in the sections “Modelling ecosystem-level energy fluxes” and “Validation of the ecosystem flux model using field data” below).”*

Finally, we now provide a clearer definition of what we mean by a population, noting that not every species was found in every stream as one might expect if these were cultured strains in a highly controlled laboratory experiment, rather than invertebrate species collected from a natural warming experiment:

Ln68: *“Note that these species were not found in every stream, but there were still multiple populations of each species, whereby a population is a unique species \times stream combination.”*

We should also note that the 20 species we mentioned in the original manuscript are actually reduced to 16 species after the quality control procedures, with four species excluded for having an $r^2 < 0.5$ or $p > 0.05$ for any term in the model. This is clearly reported in Table S5 and we apologise for any confusion caused.

Currently, the statistical modelling treats species as the sole random effect. It is likely that populations within the same streams, or even between streams and despite temperature differences, share a geographic/spatial autocorrelation. Demonstrating that this spatial component to the data is not an issue, or capturing it, is probably necessary.

Response #4: We now include a supplementary analysis where we explore potential effects of spatial autocorrelation in the data on our results. This includes a Mantel test of pairwise temperature differences and pairwise distances between streams, which shows that streams of similar temperature are not significantly grouped in the landscape (Mantel $r = -0.1293$, $p = 0.780$). We also present a semivariogram of the residuals from our best-fitting model, which shows no evidence of patterns in the residuals as a function of pairwise distance between sampling points (Fig. S3). A Moran’s I test also indicates that there is no significant spatial autocorrelation in the model residuals (Moran’s $I = 0.1187$, $p = 0.453$). Please see the “Exploration of spatial autocorrelation” section in Supplementary Information for more

details. We also added a sentence to the main text to acknowledge this finding and direct readers to the relevant supplementary information:

Ln87: *“Supplementary Analyses indicated no influence of spatial autocorrelation on these results, i.e., warm and cold streams are sufficiently mixed in the landscape (Fig. S3).”*

Furthermore, it is perhaps now best practice to perform analyses of allometric relationships in the context of phylogenetic information, not just species identity. Given that the authors have full details on the genus and species of the 20 invertebrates, one might expect full use of a phylogeny (easily crafted from databases). Even if this were not possible, inclusion of a nested hierarchy of taxonomy would be a stop-gap.

Response #5: We agree that this is an important addition to make, though it was far from easily crafted! We have recruited an additional co-author Dimitrios-Georgios Kontopoulos, with the relevant specialist expertise. Our new analysis is presented in the *“Exploration of phylogenetic structure”* section in Supplementary Information. In short, we first reconstructed a phylogeny by combining tree topology information from the Open Tree of Life (Table S2), COI-5P nucleotide sequences, and age estimates from the TimeTree database (Fig. S4). We next fitted a series of alternative phylogenetic and non-phylogenetic regression models. We found that accounting for evolutionary relatedness did not qualitatively alter our previous results, with the optimal model from this exercise (Table S3) containing exactly the same terms as those presented in Table 1a. A visualisation of the significant interaction terms in the phylogenetic model (Fig. S5) also reveals a remarkably similar result to that presented in Fig. 1. Therefore, the main conclusions of our study remain unaffected, which we acknowledge in the main text:

Ln89: *“In addition, the optimal model describing our data (Table 1a) remained the same after accounting for phylogenetic information, suggesting that both evolutionary and acclimatory processes play important roles in shaping metabolic versatility (Table S2-S3; Fig. S4-S5; see Supplementary Analyses).”*

Overall, the MS would benefit from a more robust statistical analyses that account for the spatial and phylogenetic structure of the data. In most analyses that do this, standard errors actually increase. One might wonder whether the results of the ANCOVA like analysis (e.g. quite subtle variation in the slopes , Fig 1) will be retained once these sources of variation are accounted for.

Response #6: We find the comment that there is “quite subtle variation in the slopes” of Fig. 1 rather harsh, and we contend it is in fact anything but “subtle”. On average, the allometric scaling exponent in Fig. 1b ranges from 0.53-0.75 (a difference of 0.22) and the activation energy in Fig. 1d ranges from 0.63-0.96 (a difference of 0.33), and this does not even include the upper and lower confidence intervals of these estimates. These are quite frankly huge deviations for exponents that are often suggested to be universally fixed at 0.75 and 0.65, respectively (see dashed lines in Fig. 1). Given the substantial variation we detected in these exponents as a function of chronic temperature exposure, it is hardly surprising that “the results of our ANCOVA like analysis” are retained following the suggested additional explorations of spatial and phylogenetic structure (see Responses #4 and #5 above).

Second, the introduction to the work highlights two sources of ‘plasticity’ - acclimation, which I interpret from the writing as short term (non-evolutionary) physiological responses of the different individuals to temperature variation - and adaptation, which I interpret from the writing as evolutionary change derived from allele frequency changes. It’s probably worth noting that in most of evolutionary biology, sources of variation

derived from adaptation are not considered plasticity, which is reserved for variation not attributable to allele frequency changes in a population. Thus, I expect readers will have trouble aligning the idea of metabolic plasticity with physiological adaptation.

Response #7: This is a valuable comment. Your interpretation of acclimation and adaptation is consistent with our own. We struggled to think of a term that signified the potential for changes in metabolism that could be due to either one or both of these processes. We appreciate that “plasticity” is more typically thought of as reversible, and so better suited to just the acclimation component. We now rebrand this as “metabolic versatility” throughout the revised manuscript to ensure the concept is better aligned with both acclimation and physiological adaptation, *e.g.*, the title is now “*Metabolic versatility can amplify ecosystem responses to global warming*”.

Given the nature of the work, question, and the sampling and measuring over a very short term of individuals, one could probably just leave the idea of adaptation out of this. However, it is worth recognising that perhaps the reason for bringing this idea into the MS is that the chronic exposure natural experiments are indicative of long term adaptation to a temperature regime. But these differences among chronic temperature are rarely defined as plasticity.

Response #8: We think it would be wrong to leave adaptation out of the story given that the study organisms are potentially exposed to their respective thermal regimes for centuries and thus many generations (*i.e.* evolutionary time scales). This is particularly true for the organisms with an entirely aquatic life stage, *e.g.* the snails and mites. Indeed, genetic differentiation among populations of the snail *Radix balthica* collected from streams of different temperature in the Hengill system has been demonstrated in previous work (Johansson *et al.* 2016 *Journal of Evolutionary Biology*, cited in the manuscript as reference #35). Our new phylogenetic analysis also confirms that both evolutionary and acclimatory forces shape the metabolic response to chronic temperature exposure (see Response #9). As noted in the previous comment, we have instead rebranded “metabolic plasticity” as “metabolic versatility” to avoid implying that the changes in metabolism are purely reversible.

Perhaps there is opportunity, however, to consider more carefully how the ‘levels of metabolic plasticity’ could be due to a combination of acclimation or adaptation. Plasticity is defined by how a group (population, size class, species) responds to two or more different environments. That this plasticity might vary with the history the ‘grouping variable’ experiences is a very well established way to ask whether evolution has shaped how plastic populations are. It strikes me that the analysis that crosses chronic and acute conditions are indeed partitioning the impacts of adaptation to a temperature regime with acute exposure to a wider variety of conditions. Articulating this quantitatively does require careful attention to the statistical modelling.

Response #9: Thank you for this insightful comment. Disentangling the influence of acclimation and adaptation on metabolic rate is indeed key for accurately forecasting biological responses to global warming. To shed light on this, we now estimate the phylogenetic heritability of metabolic rate measurements using the phylogeny that we reconstructed (see Fig. S4 and Response #5 above). The resulting phylogenetic heritability estimate was 0.48, which indicates that nearly half of the variation in metabolic rate can be explained by its gradual evolution across species. In other words, closely-related species tend to have more similar metabolic rate values than randomly selected species. When we additionally controlled for body mass, acute temperature exposure, and chronic temperature exposure, the unexplained variation in metabolic rate decreased from 52% to 8%. All these results combined suggest that

evolutionary and acclimatory processes play important roles in shaping metabolic versatility across the species and environments in our study. Please see the new “*Exploration of phylogenetic structure*” section in Supplementary Information for more details.

Third, it is not clear in the MS just why evidence for metabolic plasticity and its effects on ecosystem function is so important to understand. The abstract leaves an impression that it is that some experiments have had community respiration rise in experiments even after treatments cease (the wording is a bit hard to understand...). Is this the major contribution of the findings in the MS?

Response #10: This is the correct interpretation, but we acknowledge that the message was not clear enough in our original manuscript. One might expect organisms to acclimate / adapt over time, such that the impacts of warming should decline in the long-term. Instead, a recent high-profile study (Yvon Durocher *et al.* 2017 *Nature Climate Change*, cited in the manuscript as reference #27) showed that ecosystem respiration increased even further after seven years of experimental pond warming relative to results after just one year of warming. The amplification of ecosystem respiration is thus surprising and should result in greater carbon emissions to the atmosphere. Our model prediction for long-term exposure to warmer conditions to amplify ecosystem fluxes (which directly correlate with ecosystem respiration; see Fig. S6) adds weight to the generality of this experimental finding, and highlights a new framework for using size-dependent metabolic versatility to predict these surprising effects. We have reworded the abstract to make it clearer that “*long-term warming amplifies ecosystem respiration rates through time in recent mesocosm experiments*” and we have also articulated the above reasoning in the revised manuscript to clarify the importance of our work:

Ln141: “*The associated increase in respiratory losses could help to reconcile the apparent paradox of amplified ecosystem respiration after seven years of warming in a pond mesocosm experiment, relative to the effects after just one year of warming²⁷. If organisms were adapting to mitigate the effects of warming over longer time scales, then ecosystem respiration should converge with the controls in this experiment. The amplification of ecosystem respiration is thus surprising and should lead to greater carbon emissions to the atmosphere²⁷. Our model predictions for amplified ecosystem fluxes following chronic exposure to warmer conditions thus highlight the value of using size-dependent metabolic versatility to predict these surprising long-term warming effects on ecosystems.*”

It is compelling that the variation reporting in the MS can account for an additional 60% of energy flux, but it’s not clear what this means: is the MS indicating that there was a lot of unexplained variation in energy flux in natural communities and that metabolic plasticity captures it? Or that that including plasticity in the modelling leads to 60% higher energy flux than previously reported?

Response #11: We meant the latter: including metabolic versatility in the modelling leads to 60% higher energy flux than predicted by the current state-of-the-art modelling framework based on ecological metabolic theory. We have clarified this in the revised abstract:

Ln25: “*A mathematical model parameterised with these findings shows that metabolic versatility could account for 60% higher ecosystem energy flux with just +2 °C of warming than a traditional model based on ecological metabolic theory.*”

Overall, the MS would thus benefit from more clarity on the gap in knowledge that the data and results fill. Of course, another framing might be ‘how does accommodating metabolic plasticity in metabolic rates among species (interspecific variation) alter our

understanding of ecosystem function’. This is perhaps more generic and leaves open the options that the persistent averaging implicit in the core allometry of the MTE, used in a wide variety of circumstances, is just fine, or it isn’t.

Response #12: Thank you for encouraging us to drill down on the important knowledge gap that our study fills. We hope our edits in relation to Responses #10 and #11 above have helped in that regard. We also added a sentence about the general implications of our new modelling framework for predicting global warming impacts on ecosystem functioning, as suggested:

Ln150: *“More generally, accounting for different rates of metabolic versatility among species could provide a better understanding of how species-specific responses to global warming may sum up to alter ecosystem functioning.”*

Finally, the MS would benefit from a gentle reminder about how narrow or general the mathematical model is. There are strong references to the development and use of the model in other papers and against other questions and some big numbers on feeding interactions and body masses thrown about. In the end, it’s a well parameterised and well established model of the system. But what features of this make it general enough to then remark on the role of metabolic plasticity in ecosystem function? I imagine that readers may not be convinced that the model parameterised to this system (very classy and well done as it is) is the most effective tool for generalising about the effects of metabolic plasticity. If the consequences were to hold up in more general model formulations - e.g. across different distribution of body sizes and or interaction distributions - the message would be very strong indeed. Perhaps a bit more care is needed here to balance the incredible opportunity for accuracy of modelling this system with the capacity to generalise beyond it.

Response #13: This is a very valid point. We now note that the size-dependent nature of the model (rather than a reliance on taxonomic identity) allows it to scale across multiple trophic levels and should make it generalisable to other ecosystems.

Ln123: *“Note that the size-dependent nature of the model makes it independent of species identity and enables it to scale across multiple trophic levels²⁶, which should make it generally applicable to other ecosystems.”*

We also agree, however, that it is worth noting the need for further validation of our modelling framework against other datasets to test the generality of our findings beyond our model system, and we incorporate this caveat into the revised discussion:

Ln176: *“Our modelling framework should also be tested against ecosystems with different distributions of body sizes and trophic interactions to test the generality of our findings beyond our focal study system.”*

Reviewer #2 (Remarks to the Author):

In this paper, the authors measure metabolic rate of stream invertebrates across a range of species, body masses, temperatures, and ‘home’ climates to evaluate the plasticity of the size- and temperature-dependence of metabolic rate. They leverage the very cool Iceland stream system to do this, since it provides a natural opportunity to evaluate the consequences of long-term exposure to temperature change. The results suggest that, yes,

the temperature you experience over long periods of time influences your response to short-term changes in temperature (metabolic plasticity). The results both contradict notions of universal temperature- and size-dependence of metabolic rate, and using a flux model, suggest that the observed plasticity could have long-term effects for the consequences of warming on ecosystem respiration. Overall I enjoyed reading the study. It was well-written and clear, and directly explores an important question with a large and compelling data set. The methods appear sound to me. The results also should be highly impactful, as they provide a compelling update to broad views on the mass- and temperature-dependence of metabolic rate.

Response #14: Thank you for this ringing endorsement of our work. We are grateful for the additional constructive feedback, which we have taken on board in our responses below.

Some small points:

L49 – Previous research where? On this system?

Response #15: Yes, we have now clarified that we meant “*previous research in the Hengill system*”.

L56 – It seems unlikely that the O₂ measurements are for ‘normal’ situations. Confined in a vial, these seem more likely to reflect resting metabolic rate rather than routine. If they could swim and flee predators and forage, maybe ‘normal’, but without that they should be thought of as closer to resting.

Response #16: We have rephrased this statement from “*normal activity*” to “*organisms exhibited some activity*” during the experiments. We also acknowledge in the revision that our metabolic rate measurements lie somewhere between resting and routine metabolic rate:

Ln194: “*Organisms in our experiments were confined to small glass vials, and thus their activity was constrained during the experiments compared to normal activity. As such, our measurements are much closer to resting than maximum metabolic rate, and temperature effects are thus likely to indicate differences in energy allocated to growth, rather than activity.*”

L68 – What do you mean by ‘capable’? Are you suggesting that changes in metabolic rate are intentional rather than biochemical? Might want to rephrase or explain this point.

Response #17: Thank you for spotting the inappropriate phrasing here. We have changed this from “*were more capable of elevating their metabolism*” to the passive “*had a more elevated metabolism*” in the revision to reflect the likelihood that this is driven by biochemical reactions at the cellular level, rather than behavioural choices of the organism.

L147 – Do you actually expect warming to influence this system, since the temperature is controlled by the geothermal energy?

Response #18: We were not specifically remarking on warming in the Hengill system, rather the general scope for increased performance at higher temperatures amongst energetically constrained organisms from a cold region. We have rephrased “*as it warms*” to “*as temperature increases*” in the revision to avoid any confusion.

Reviewer #3 (Remarks to the Author):

Line 16-17: I think the statement “empirical evidence for this metabolic plasticity across species within food webs is lacking” is ambiguous as it could be interpreted as meaning that we know little about metabolic plasticity, which is not the case. There has been considerable research on metabolic plasticity. For instance, in ectothermic animals alone, Seebacher et al. (2015) collated 637 measurements of 202 species of the thermal acclimation responses of physiological traits including metabolic rates, heart rates, enzyme activities and locomotor performance. And presumably the animal species within this dataset occupy different levels within food webs. I would suggest rephrasing this statement to something like “the consequences of among-species variation in metabolic plasticity for energy flow through food webs is currently unknown”.

Response #19: Thank you for the recommendation for a more transparent statement on the current gap in the field relevant to our manuscript. We have rephrased the opening sentence of the abstract as recommended.

Line 20-21: I think rephrasing this sentence to something like “Chronic exposure to higher temperatures also increases the acute thermal sensitivity of whole-organismal metabolic rate independent of body size” would improve clarity.

Response #20: Done.

Line 36-37: I think Seebacher et al. (2015) should be cited here as well.

Response #21: We agree that this is an important paper to cite and have added it to the revision.

Line 54: The streams also seem to vary considerably in the extent to which they vary seasonally. For instance, IS10, IS13 and IS8 have relatively stable thermal regimes ranging from 3.4-19.4°C, whereas IS12 and IS1 show dramatic seasonal fluctuations in temperature. Have you considered including a measure of seasonal variability into your models to determine if this also contributes to variation in the acute thermal sensitivity of metabolic rate or to variation in the scaling of MR with body size? This would be particularly interesting in the context of the theory of metabolic cold adaptation (MCA), which suggests that organisms living in colder climates (such as those high latitudes and altitudes) should have higher metabolic rates than those living in warmer climates when measured at the same temperature. Some comparative analyses that examine how metabolic rate varies along geographic gradients (such as latitudinal clines) have found support for the MCA hypothesis (e.g. Addo-Bediako et al. 2002), but such analyses are unable to determine whether this observed pattern is the result of variation in mean annual temperature or variation in seasonality (see Alton et al. 2017).

Response #22: This is a keen observation, though we would point out that there are examples of both cold and warm streams that have high variability (e.g. IS12 and IS2, respectively) and low variability (e.g. IS13 and IS8, respectively). As such, temperature variability is unlikely to have a significant influence on our findings. To test this more formally, we accounted for stream temperature variability by conducting a supplementary analysis where we included the standard deviation of chronic temperature exposure as a random slope in our modelling framework. This had no qualitative effect on our main conclusion, with the optimal model describing our data (Table S4) containing all the same terms as those presented in Table 1a. A visualisation of the significant interaction terms in this new model (Fig. S7) also reveals a

remarkably similar result to that presented in Fig. 1. We have included some text on this new analysis when we discuss the study system in our Methods section:

Ln325: *“The streams exhibit some differences in the annual variability of their thermal regimes, but there are examples of both cold and warm streams that have high (IS12 and IS2) and low (IS13 and IS8) variability throughout the year. Our main finding is also robust to the inclusion of stream temperature variability as a random effect in our modelling framework (Table S4; Fig. S7).”*

Line 69-77: I think it is important that you provide some discussion about the limitations of the experimental component of your research as this would provide some valuable guidance for future research. In particular, I think it would be valuable if you acknowledged that you did not measure activity levels during your metabolic rate measurements and therefore cannot account for this important source of variation in estimates of metabolic rate (see Halsey et al. 2015). By not conducting simultaneous measures of activity and metabolic rate you cannot disentangle changes in activity from changes in mass-independent resting metabolic rates (which should provide a better estimate of the cost of self-maintenance), and thus you are unable to determine the relative contributions of behavioural plasticity and physiological plasticity to observed changes in rates of energy expenditure. While you say that “higher metabolic rates can provide organisms with greater scope for faster growth”, this would only be the case if it was resting metabolic rates that increased (and only if resting metabolic rates are indicative of the capacity of individuals to process food and positively correlated with fitness as per the increased-intake hypothesis, see Nilsson 2002). If it was increases in activity levels that was causing routine metabolic rates to increase, then this would suggest that more energy was being allocated to activity, which may reduce the amount of energy available for growth. That you did not measure activity is not intended to be criticism – in my opinion, your measures of metabolic rate in this system are an incredibly valuable contribution to the field of research dedicated to understanding the influence of temperature on metabolic rates. I do however think that an acknowledgement of this limitation only adds value to your contribution because it should help to inspire future research directions.

Response #23: This is an excellent point, which we have incorporated into our revision through the addition of a new paragraph of discussion:

Ln193: *“Activity levels can be an important source of variation in estimates of temperature effects on metabolic rate²⁹. Organisms in our experiments were confined to small glass vials, and thus their activity was constrained during the experiments compared to normal activity. As such, our measurements are much closer to resting than maximum metabolic rate, and temperature effects are thus likely to indicate differences in energy allocated to growth, rather than activity. Nevertheless, quantification of activity levels and/or measurement of resting and maximum metabolic rates would be needed to disentangle the relative contributions of behavioural and physiological versatility to the observed changes in rates of energy expenditure. Follow-up studies should prioritise this research gap.”*

Line 98, Table 1: Change “9 streams” to “nine streams”.

Response #24: Done!

Line 112: Your experimental data provides a measure of metabolic plasticity in invertebrates that you say are the dominant primary and secondary consumers in the

streams. Did you assume that the same level of plasticity across all trophic levels or just within the trophic levels that you have data for?

Response #25: The model is independent of species identity and trophic levels, but does capture the latter through the size-dependence of metabolic plasticity (which decreases with increasing body size). Given that trophic level increases with body size, then the capacity for elevated metabolism should decline as you move further up the food web. We have added a sentence to the revised manuscript noting that the model should be generally applicable to different trophic structures, citing Riede *et al.* (2011) in *Ecology Letters*, which demonstrated a positive relationship between body size and trophic level across aquatic and terrestrial ecosystems:

Ln123: “*Note that the size-dependent nature of the model makes it independent of species identity and enables it to scale across multiple trophic levels²⁶, which should make it generally applicable to other ecosystems.*”

Line 268-269: Were these experiments conducted throughout the year or only during particular seasons?

Response #26: We realise our original wording was imprecise here. We have rephrased this sentence in the revision:

Ln318: “*Fieldwork was performed in the summers of 2015-2018, between May and July.*”

Line 278-282: Was the water from the main river filtered?

Response #27: We note in the revision that “*The water was passed through a 125 μm sieve to ensure no organisms or filamentous algae entered the aquaria*”.

Was the home-stream temperature set to the average stream temperature or the temperature at the time of collection?

Response #28: We now state that the “*chambers were set to the home-stream temperature of the organisms during sampling*”.

How do you know that your organisms were not feeding while they were being held in the lab? If they were being housed together in groups, they may have eaten each other, or if individuals died then those that were living may have fed on the carcasses. It is entirely reasonable in such cases that you would not know the digestive states of the individuals and therefore just have to accept that this is source of variance in your data that you cannot account for.

Response #29: We know the organisms were not feeding while they were being held in the lab because the aforementioned filtering of water used in the aquaria was aimed at “*limiting the potential food available to the study organisms*”. Nevertheless, we do acknowledge that small organic particles could have entered the aquaria through the gaps in our 125 μm mesh sieve, or clinging to the bodies of the collected organisms, and so we cannot completely rule out the possibility of some feeding prior to experiments. We acknowledge this and the associated unexplained variance in the revision:

Ln345: “*While we did not observe any cannibalism or organisms feeding on dead bodies in the laboratory, we cannot rule out the possibility that organisms fed on fine algal or detrital particles in the water, thus increasing variability in our metabolic measurements due to differences in digestive state.*”

Line 294: Is the acclimation period you mention here the same 15-minute period over which temperature was changed?

Response #30: Correct. We have clarified that this was the “*15-minute acclimatisation period*” in the revision.

Line 298-299: You mention on Line 57 that measurements were for less than 1 h, but what was the minimum measurement duration?

Response #31: We now clarify that our acute exposures involved “*experiments lasting 10-60 minutes*”.

Line 301-302: What were the chamber volumes? What was your cleaning protocols for the chambers and what were your calibration protocols for the oxygen sensors? I would highly recommend consulting Killen et al. (2021) who provide a checklist of criteria for reporting aquatic respirometry methods. While this checklist is for intermittent-flow respirometry, some of the criteria are also relevant for closed-system respirometry. This level of detail can be presented in the supplementary material.

Response #32: We now specify that the “*chambers ranged in volume from 0.8–5 ml and scaled with the size of the organism*” and that this volume was “*equal to 5–100 times the mass of the measured organism*”. We note that “*the system was cleaned with bleach at the end of each measurement day to avoid accumulation of microbial organisms on the insides of glass chambers and the water bath*”. We have scrutinised the Killen *et al.* (2021) checklist and think the other key points that are relevant to closed-system respirometry are included in our methods or supplementary information.

Tables 1 and S3-S5: In your tables you have used the abbreviations of TS and TL but have not explained what these mean in the Table legends. Is TS stream temperature and thus equivalent to chronic temperature, TC? I do not know what TL would stand for?

Response #33: Apologies, this was a relic of our initially labelling them as short- and long-term temperature exposures (T_S and T_L , respectively). We have corrected the abbreviations in all tables to T_A and T_C and note that we already define them as acute and chronic temperature exposures, respectively, in the table legends.

Reviewers' Comments:

Reviewer #1:

Remarks to the Author:

Thank you for a very thorough set of responses and new analyses. These have increased the clarity of the inference against revised and more compelling statements of the gaps in knowledge. I am impressed by the efforts overall.

I am not totally convinced that the new term versatility helps, but it's an interesting term to use to try and capture what you mean by variation at the scales you are reporting on.

I was actually expecting, simply, more of a definition of plasticity and acclimation in the work to align with the sources of variation (or not).

Reversibility is not a distinguishing feature of plasticity. As indicated in the review, plasticity is defined by a group of similar organisms responding differently when the environment changes. Acclimation is different to this.... it's the re-gaining of a previous or modified state after an acute shift in state. I'd be more inclined to see plasticity retained, but just better defined in the context of the MS.

Reviewer #3:

Remarks to the Author:

Thank you for adequately addressing my comments. I have no further comments.

*** Note that original reviewers' comments are in bold. Our responses are numbered sequentially, written in plain font, and contain quotations to the revised text in italics with line numbers corresponding to the revised version of the manuscript ***

Reviewer #1 (Remarks to the Author):

Thank you for a very thorough set of responses and new analyses. These have increased the clarity of the inference against revised and more compelling statements of the gaps in knowledge. I am impressed by the efforts overall.

Response #1: Thank you for recognising the effort we put into addressing your original comments, which helped to greatly improve our manuscript. We are happy that you are satisfied with the revised version.

I am not totally convinced that the new term versatility helps, but it's an interesting term to use to try and capture what you mean by variation at the scales you are reporting on. I was actually expecting, simply, more of a definition of plasticity and acclimation in the work to align with the sources of variation (or not). Reversibility is not a distinguishing feature of plasticity. As indicated in the review, plasticity is defined by a group of similar organisms responding differently when the environment changes. Acclimation is different to this.... it's the re-gaining of a previous or modified state after an acute shift in state. I'd be more inclined to see plasticity retained, but just better defined in the context of the MS.

Response #2: We agree with the reviewer and have reinstated the “metabolic plasticity” terminology throughout. We have also provided a clearer definition of the phrase at first mention in the introduction:

Ln42: “We refer to this flexibility in species-level thermal responses henceforth as “metabolic plasticity”, which can be thought of simply as a group of similar organisms altering their metabolic rate in a similar way when the environment changes.”

Reviewer #2 (Remarks to the Author):

Thank you for adequately addressing my comments. I have no further comments.

Response #3: We are very grateful for your constructive criticism on the original manuscript and pleased that you have approved our revision.